# Low FXIII activity levels in patients with vascular anomalies, not in patients with trauma-induced coagulopathy

Hiroaki Otsubo[1], Tsuguaki Terashima[2], Mai Terashima[1], Siqiang Gao[3], Mika Ogawa[1], Hidefumi Kato[3], Kyosuke Takeshita[4], Shohei Mizuno[5], Satsuki Murakami[5], Akiyoshi Takakmi[5], Katsuhiko Matsuyama[6], Hiroshi Furukawa[7], Eizo Watanabe[2], Takayuki Nakayama[1]*

1 Department of Clinical Laboratory, Aichi Medical University, Nagakute, Aichi, Japan, 2 Department of Emergency and Critical Care Medicine, Aichi Medical University, Nagakute, Aichi, Japan, 3 Department of Blood Transfusion, Aichi Medical University, Nagakute, Aichi, Japan, 4 Department of Clinical Laboratory, Saitama Medical Center, Kawagoe, Saitama, Japan, 5 Department of Hematology, Aichi Medical University, Nagakute, Aichi, Japan, 6 Department of Cardiac Surgery, Aichi Medical University, Nagakute, Aichi, Japan, 7 Department of Plastic and Reconstructive Surgery, Aichi Medical University, Nagakute, Aichi, Japan

* tnaka@aichi-med-u.ac.jp

## Abstract

Patients with localized intravascular coagulopathy (LIC) related to vascular anomalies occasionally suffer from hemorrhage and delayed wound healing, presumably due to consumptive reduction of fibrinogen and platelets. However, LIC patients with normal levels of platelets and fibrinogen can be hemorrhagic for unknown reasons. These facts led us to hypothesize that factor XIII (FXIII) is decreased since FXIII plays an important role in hemostasis, tissue repair, and wound healing. We collected plasma from 17 patients with LIC (9 with aneurysm, 8 with vessel malformation) and 7 patients with trauma-induced coagulopathy (TIC), which showed a fibrinolytic phenotype similar to LIC, and measured FXIII levels. FXIII levels were considerably decreased (7–53%, mean 29.4%) in LIC patients. 13 of 17 patients showed bleeding tendency even though DIC scores were low. However, FXIII levels were only slightly decreased (39–98%, mean 67.4%) in TIC patients. We also investigated the correlation between FXIII activity and coagulation parameters. FXIII activity most strongly correlated with fibrinogen content (R=0.882, P<0.001) in LIC patients. FXIII levels below 50% were observed when fibrinogen was less than 200 mg/dL. Based on these findings, we successfully treated two LIC patients with FXIII concentrates. Decreased FXIII activity was specific to LIC, not TIC. Testing for FXIII activity should be considered in LIC patients even with normal fibrinogen levels, who are scheduled to undergo any invasive procedure, since FXIII supplementation is occasionally required for them.

**Data availability statement:** All relevant data are within the manuscript.

**Funding:** JSPS KAKENHI JB25K11657.

**Competing interests:** The authors have declared that no competing interests exist.

## Introduction

Disseminated intravascular coagulation (DIC) is a crippling coagulopathy that can occur in critically ill patients with a variety of disorders, such as infections and malignant diseases. DIC is characterized by organ failure due to systemic microvascular thrombosis, and bleeding tendency caused by the consumption of coagulation factors [1]. Based on fibrinolytic activity in the plasma, DIC is classified into three subtypes: enhanced-fibrinolytic, balanced-fibrinolytic, and suppressed-fibrinolytic DIC [2]. In most cases, DIC occurs suddenly and requires urgent treatment. However, vascular anomalies, such as aortic aneurysms and vascular malformations, can lead to localized intravascular coagulopathy (LIC) [3], which can persist over time and gradually develop into DIC in some cases. This gradual development of DIC is called chronic DIC (cDIC) [4]. Patients with LIC frequently suffer from hemorrhage and delayed wound healing, presumably due to a consumptive reduction of fibrinogen and platelets because LIC is hyperfibrinolytic [4]. However, LIC patients with normal levels of platelets and fibrinogen can be hemorrhagic for unknown reasons according to clinical observations among physicians at our facility.

Factor XIII (FXIII) FXIII is a Ca(2)+-dependent enzyme which forms covalent -(gamma-glutamyl)lysine cross-links between the gamma-carboxy-amine group of a glutamine residue and the -amino group of a lysine residue. Substrates of FXIIIa are diverse [5]. Among them, the action of FXIII on fibrin and extracellular matrix proteins, such as fibronectin, collagen and von Willebrand factor are especially important. FXIII stabilize fibrin clots able to withstand mechanical and enzymatic breakdown through fibrin-fibrin cross-linking and cross-linking of fibrinolysis inhibitors, in particular alpha2-antiplasmin. FXIII also promotes tissue repair and wound healing via cross-linking of extracellular matrix proteins [6]. Thus, decreased activity of FXIII in the plasma induces bleeding tendency and delayed wound healing [7]. According to the Human Gene Mutation Database, approximately 200 mutations in the FXIII gene cause congenital FXIII deficiency (https://www.hgmd.cf.ac.uk/ac/search.php). FXIII supplementation for prophylaxis in patients with congenital FXIII deficiency is established [8,9]. Additionally, various conditions such as sepsis, multiple fractures, and DIC decrease the FXIII content in plasma [10]. However, little is known about which subtype of DIC is associated with acquired FXIII deficiency. The threshold of FXIII level in the plasma to be replenished by FXIII concentrates in patients with acquired FXIII deficiency has not yet been established as well [11]. Whereas the risk of spontaneous bleeding in congenital FXIII deficiency increases below ~10–15% activity of FXIII [12].

These facts led us to survey the activity of FXIII, which can be involved in delayed wound healing and bleeding tendency, in the plasma of patients with LIC, and we found that the specifically decreased Factor XIII could be involved in the bleeding tendency of LIC patients. Moreover, we describe two cases where FXIII replacement may have contributed to hemostasis in LIC patients.

## Patients and methods

Between July 2017 and March 2023, clinicians retrospectively identified patients with vascular anomalies accompanied by persistently increased FDP (> 20 μg/

mL for more than 3 months). Patients with concomitant diseases that may induce coagulopathy were excluded. Patients with traumatic injuries which required hospitalization for intensive care were served as a comparison group. Clinical records were accessed for research purposes between October 3, 2024, and December 11, 2024. This study was approved by the Research and Ethics Committee of Aichi Medical University (ID number of the approval: 2024-154).

### DIC scoring

DIC score was calculated using the ISTH (International Society on Thrombosis and Haemostasis) [13] and JMHW (Japanese Ministry of Health & Welfare) [2] DIC scoring systems. The DIC scores in each patient were confirmed in several time points.

### Blood testing

Standard clinical laboratory tests were performed to assess hemostasis right after clinicians found vascular anomalies or traumatic injuries. Complete blood cell counts and coagulation markers such as fibrinogen, prothrombin time, activated partial thromboplastin time, fibrin/fibrinogen degradation products (FDP) and D-dimer were quantified using automated analyzers, XN-1000 (Sysmex, Kobe, Japan) and CP3000TM® (Sekisui Medical, Co. Inc., Tokyo, Japan), respectively. Plasma samples (not frozen) were sent on the day of blood draw to a commercial laboratory (BML Inc., Tokyo, Japan) for assessment of FXIII activity measured by colorimetric synthetic substrate method. Briefly, FXIII in the plasma was activated by thrombin in the reagent, and then activated FXIII reacted with the synthetic substrate to release ammonia. Free ammonia converted NADH (reduced nicotinamide adenine dinucleotide) to NAD+ (oxidized form), and the rate of decrease in absorbance at 340 nm during this process is measured. Since the rate of change in absorbance is proportional to the FXIII activity in the sample, the FXIII activity value is calculated using a standard curve. FXIII activities and coagulation markers in each patient were confirmed in several time points. Some patients also underwent measurements of von Willebrand factor (vWF) activity. The plasma samples were sent on the day of blood draw to a commercial laboratory (BML Inc., Tokyo, Japan) for assessment of vWF ristocetin cofactor activity.

### Statistical analysis

The Pearson correlation coefficient between FXIII activity and other coagulation markers was calculated using the R statistical computing software (version 4.2.2; The R Foundation for Statistical Computing, Vienna, Austria). Data normality was tested by the Shapiro-Wilk test with the SPSS Statistics version 17.0 software (SPSS Japan Inc., Tokyo, Japan). Statistical significance of group differences was evaluated by the Student $t$ test using Excel software (Microsoft, Redmond, WA). P values < 0.05 and ≥0.1 were considered statistically significant and not statistically significant, respectively. When P values were marginally significant (≥0.05 to <0.1), we described P values.

## Results and Case presentations

### FXIII activity is critically decreased in patients with LIC, but not with TIC

DIC scores, FXIII activities and coagulation markers in each patient were confirmed in several time points, and data at the initial visit were shown in Tables 1 and 2. All 17 patients with vascular anomalies exhibited chronic (for more than 3 months) coagulopathy (FDP > 20 μg/ml), and 13 out of 17 patients showed a bleeding tendency such as epistaxis, cutaneous bruising, continuous bleeding from wounds, and oral cavity bleeding. However, only 5 and 3 patients met the criteria for the JMHW DIC score and the ISTH DIC score, respectively. We found that the FXIII levels in plasma obtained from all 17 patients were mildly to severely decreased (Table 1; mean 33.1%, range 7–59%).

**Table 1. Clinical and laboratory profiles of patients with localized intravascular coagulopathy.**

| Patient | Underlying disease | Bleeding tendency | | PT ratio | PT(INR) | APTT [ref] (sec) | Plt (×104/µL) | Fibrino-gen (mg/dL) | FDP (µg/mL) | D-dimer (µg/mL) | F XIII (%) | DIC score (JMHW) | DIC score (ISTH) |
|---|---|---|---|---|---|---|---|---|---|---|---|---|---|
| | | | Normal range | – | 0.90-1.10 | | 18.0-35.0 | 190-330 | <5.0 | <1.0 | 70-140 | | |
| 1 | TAA | Ecchymosis | | 1.18 | 1.26 | 27.8 [26.3] | 5.2 | 88 | 191.1 | 89.73 | 28 | 8 | 4 |
| 2 | Kasabach-Merritt synd | None | | 1.15 | 1.28 | 33.9 [27.2] | 2.6 | 64 | 59.5 | 28.43 | 24 | 8 | 4 |
| 3 | Venous malformation | Ecchymosis | | 1.11 | 1.06 | 25.3 [27.3] | 16.7 | 138 | 91.7 | 44.42 | 33 | 4 | 3 |
| 4 | Venous malformation | Subdural hematoma after delivery | | 1.19 | 1.33 | 35.6 [27.2] | 13.7 | 75 | 116 | 52.77 | 18 | 6 | 5 |
| 5 | AAA | Ecchymosis | | 1.08 | 1.21 | 37.9 [27.2] | 3.4 | 118 | 54.4 | 16.86 | 25 | 7 | 3 |
| 6 | AAA | Hematuria | | 1.1 | 1.15 | 29.8 [26.3] | 11.6 | 160 | 47.1 | 16.26 | 52 | 4 | 3 |
| 7 | Lymphangioma | Hematuria | | 0.99 | 0.98 | 32.7 [28.4] | 13.9 | 228 | 74.6 | 35.33 | 53 | 3 | 3 |
| 8 | AAA | Nasal bleeding | | 1.19 | 1.28 | 37.1 [28.2] | 6.5 | 132 | 103.5 | 30.76 | 34 | 6 | 3 |
| 9 | Kasabach-Merritt synd | Hemarthrosis | | 2.32 | 2.32 | 43.3 [27.3] | 9.4 | 50 | 115.3 | 50 | 7 | 16 | 6 |
| 10 | Venous malformation | None | | 1.14 | 1.15 | 35.8 [27.5] | 16.5 | 160 | 39.4 | 14.6 | 43 | 2 | 3 |
| 11 | TAA | Chronic subdural hematoma | | 1.22 | 1.2 | 37.0 [26.7] | 15 | 231 | 80.2 | 33.12 | 41 | 3 | 3 |
| 12 | Klippel-Trenaunay synd | Excessive intraoperative bleeding | | 1.49 | 1.49 | 39.1 [26.9] | 8.3 | 51 | 248.3 | 84.17 | 9 | 7 | 5 |
| 13 | TAA & AAA | None | | 1.1 | 1.03 | 39.8 [27.3] | 16.4 | 301 | 47.2 | 20.73 | 59 | 3 | 3 |
| 14 | AAA | Postoperative hemorrhage | | 0.93 | 0.92 | 33.4 [28.9] | 9.5 | 223 | 63.3 | 18.45 | 39 | 4 | 3 |
| 15 | Venous malformation | None | | 1.1 | 1.08 | 29.6 [27.9] | 25.4 | 249 | 21.1 | 10.03 | 53 | 4 | 3 |
| 16 | TAA | Ecchymosis | | 1.23 | 1.22 | 33.5 [28.5] | 7.3 | 122 | 146.6 | 46.52 | 14 | 6 | 3 |
| 17 | TAA | Ecchymosis | | 1.06 | 1.02 | 36.0 [28.3] | 6.8 | 139 | 61.2 | 22.27 | 30 | 6 | 3 |

Abbreviations: TAA, thoracic aortic aneurysm; AAA, abdominal aortic aneurysm; PT, prothrombin time; APTT, activated partial thromboplastin time; Plt, platelet; FDP, Fibrin/fibrinogen degradation products.

≧7 or ≧5 is diagnosed as DIC according to the Japanese Ministry of Health and Welfare (JMHW) or the International Society of Thrombosis and Haemostasis (ISTH) scoring systems, respectively. Normal ranges were shown under each assessed lab value.

Next, we evaluated FXIII levels in 7 patients with trauma-induced coagulopathy (TIC), which showed a fibrinolytic phenotype (PIC≧10 µg/ml and high FDP/D-dimer ratio) similar to LIC. However, contrary to our expectations, FXIII levels were only slightly decreased (Table 2; mean 67.4%, range 39–98%).

After we verified the normality in the LIC subgroup (P=0.7508) and TIC subgroup (P=0.6285) by the Shapiro-Wilk test, we evaluated the statistical significance of group differences with by the Student $t$ test, showing that FXIII levels in LIC patients were significantly lower than those in TIC patients (P<0.05). We also evaluated von Willebrand factor activity in 11 out of 17 LIC patients and found that they were within normal limits (mean 156.5%, range 63–300%).

**Table 2. Clinical and laboratory profiles of patients with trauma-induced coagulopathy.**

| patient | Major site of Injury | | PT ratio | PT(INR) | APTT [ref] (sec) | Plt (×104/μL) | Fibrinogen (mg/dL) | FDP (μg/mL) | D-dimer (μg/mL) | F XIII (%) | DIC score (JMHW) | DIC score (ISTH) |
|---|---|---|---|---|---|---|---|---|---|---|---|---|
| | | Normal range | 0.90-1.10 | | | 18.0-35.0 | 190-330 | < 5.0 | < 1.0 | 70-140 | | |
| 1 | Pelvic fracture | | 1.13 | 1.09 | 29.3 [27.3] | 16.4 | 297 | 210 | 124.7 | 56 | 4 | 3 |
| 2 | Subdural and subarach-noidal hemorrhage | | 1.03 | 0.98 | 25.6 [27.3] | 16.4 | 191 | 54.6 | 30.17 | 98 | 3 | 3 |
| 3 | blunt thoracic aortic injury | | 1.84 | 1.78 | 42.3 [27.3] | 11.7 | 132 | 327 | 182.6 | 39 | 7 | 5 |
| 4 | atonic bleeding | | 1.25 | 1.43 | 33.6 [27.3] | 13.9 | 222 | 21.6 | 7.59 | 59 | 3 | 3 |
| 5 | Multiple traffic injuries | | 1.29 | 1.43 | 35.5 [28.1] | 10.7 | 161 | 301.5 | 166.7 | 55 | 5 | 4 |
| 6 | clavicle and rib fracture | | 1.07 | 1.09 | 25.1 [28.1] | 26.7 | 242 | 209.9 | 118.5 | 88 | 3 | 3 |
| 7 | Multiple traffic injuries | | 1 | 1 | 25.3 [27.3] | 14.7 | 235 | 149.9 | 87.6 | 77 | 3 | 3 |

Abbreviations: PT, prothrombin time; APTT, activated partial thromboplastin time; Plt, platelet; FDP, Fibrin/fibrinogen degradation products.

≧7 or ≧5 is diagnosed as DIC according to the Japanese Ministry of Health and Welfare (JMHW) or the International Society of Thrombosis and Haemostasis (ISTH) scoring systems, respectively. Normal ranges were shown under each assessed lab value.

### FXIII activity parallels the fibrinogen level

We also investigated the correlation between FXIII activity and coagulation parameters in LIC patients, and found that FXIII activity most strongly correlated with fibrinogen level (R=0.882, $P < 0.001$) compared to platelet counts (R=0.589, $P < 0.05$), APTT (R=−0.33, $P > 0.05$), PT-INR (R=−0.63, $P < 0.05$), FDP (R=−0.678, $P < 0.05$) or D-dimer (R=−0.616, $P < 0.05$), as shown in Fig 1. FXIII levels below 50% were observed when fibrinogen was less than 200 mg/dL.

In TIC patients, FXIII activity showed a significant negative correlation with APTT (R=−0.884, $P < 0.05$) and PT-INR (R=−0.798, $P < 0.05$). No significant correlations were found with fibrinogen (R=0.297, $P > 0.1$), platelet counts (R=0.124, $P > 0.1$), FDP (R=−0.568, $P > 0.1$), or D-dimer (R=−0.555, $P > 0.1$).

### Case presentations

Case 1. A 17-year-old Japanese male patient (No. 12 in Table 1), with vast hemangioma and deformity in the left legs due to Klippel-Trenaunay syndrome, fractured his left femur in a fall. He was hospitalized for elective surgery to fracture reduction and fixation. Laboratory tests at admission revealed that he had coagulopathy characteristics attributable to hyperfibrinolytic DIC with decreased FXIII activity (9%). Anticoagulant therapy using a serine protease inhibitor (nafamostat mesylate; Torii Pharmaceutical Co. Ltd., Tokyo, Japan) was started and continued until 2 h before the surgery. Three units of cryoprecipitate (1 unit=1200 ml fresh frozen plasma) was administered one day before the surgery. However, FXIII activity was found to be slightly increased (20%) afterward. During the surgery, excessive intraoperative bleeding occurred, and three grams of fibrinogen concentrate (Fibrinogen HT, Japan Blood Products Organization, Tokyo, Japan) were administered. But the bleeding persisted. Next, 20 mL of FXIII concentrate (Fibrogammin® P, CSL Behring GmbH, Marburg, Germany, 26 unit/kg) were administered, and bleeding was fairly well controlled. The total amount of bleeding after completion of the surgery was 1141 mL, which exceeded the expected bleeding amount of 500 mL. FXIII activity was 46% the day after the surgery.

Case 2. A 77-year-old Japanese male patient (No. 11 in Table 1) had fell and developed a chronic subdural hematoma. He was referred to the hematology department because of the recurrence of chronic subdural hemorrhage after surgical evacuation. He had thoracic aneurysm accompanied by LIC with decreased FXIII activity (41%). Anti-fibrinolytic therapy with a small amount tranexamic acid (150 mg/day) was started based on a previous report [14]. However, dysarthria and gait imbalance emerged due to enlargement of a subdural hematoma.

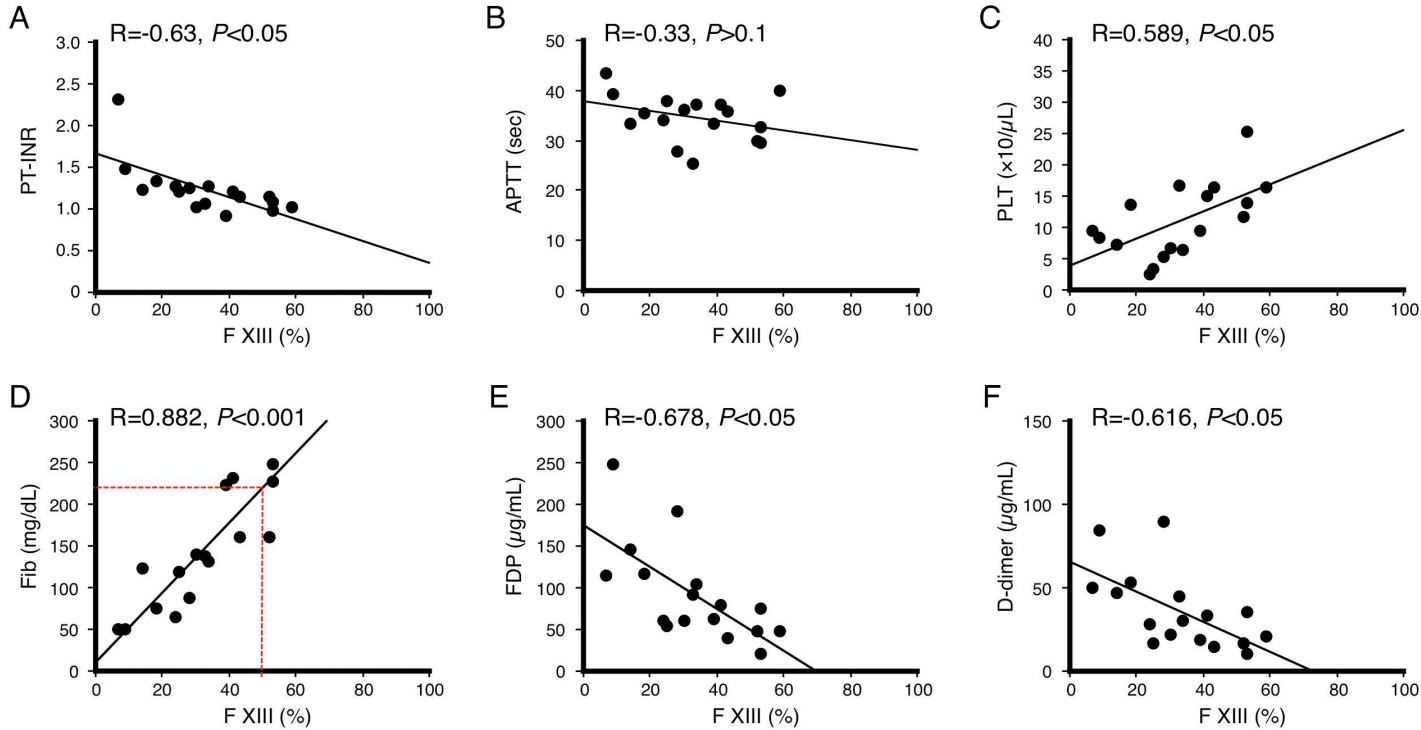

**Fig 1. Correlation between FXIII activity and coagulation parameters in patients with LIC (n = 17).** Pearson's correlation coefficient **(r)**: **A.** PT-INR (R = −0.63), **B.** APTT (R = −0.33), **C.** platelets (R = 0.589), **D.** fibrinogen (R = 0.882), **E.** fibrin/fibrinogen degradation products (FDP) (R = −0.678), **F.** D-dimer (R = −0.616).

Neurological symptoms improved with increased FXIII activity (76%) after administration of 20 mL of FXIII concentrate (Fibrogammin® P, 20 units/kg) each day for 3 days. The patient was discharged 4 days after the last administration of FXIII concentrate.

## Discussion

Reduction of FXIII in patients with LIC/cDIC has not been fully evaluated. A few case reports have suggested a putative relationship between FXIII and cDIC [15]. The reduction of FXIII was not referred to even in a report analyzing the laboratory findings from 235 patients with LIC caused by aortic aneurysm [16]. Thus, to the best of our knowledge, this is the first retrospective report to evaluate the FXIII content in plasma obtained from patients with vascular anomalies and traumatic injuries. Kleber et al. suggested that a cut-off for FXIII activity of < 50–70% may be appropriate to diagnose acquired FXIII deficiency; however, this range is very broad, opinions and diagnostic protocols may vary [17]. Here, we showed that FXIII was decreased in some cases with vascular anomalies (Table 1; mean 33.1%, range 7–59%), whereas acquired FXIII deficiency was not present in TIC patients (Table 2; mean 67.4%, range 39–98%), consistent with previous reports [17,18]. Song et al. analyzed FXIII activity in 80 patients with acute DIC and found that FXIII activity was significantly, but mildly decreased in overt DIC patients (mean 75.1%) [19]. Our data and the results from previous studies suggest that a reduction of FXIII is specific to LIC.

The sensitivity of the JMHW DIC scoring system for assessing DIC has been reported to be higher than that of the ISTH DIC scoring system [20–22]. However, only 5 patients in the present study met the JMHW DIC criteria, even though 13 of 17 patients showed a bleeding tendency, suggesting that diagnostic criteria specifically for LIC should be established.

We also investigated the correlation between FXIII activity and coagulation parameters to infer FXIII decrease from other parameters since few facilities can measure FXIII inside the hospital. We found that the FXIII levels most strongly correlated with the fibrinogen content (R = 0.882, P < 0.001) in LIC patients. FXIII levels below 50% were observed when fibrinogen was less than 200 mg/dL (Fig 1). In TIC patients, FXIII levels did not correlate with fibrinogen levels and were most strongly correlated with APTT (R = −0.884, P < 0.05) and PT-INR (R = −0.798, P < 0.05), as described above. These findings suggest that LIC and TIC may exhibit similar fibrinolytic phenotypes; however, the underlying pathogenic mechanisms are quite different.

There is currently no clear answer why FXIII levels most strongly correlated with the fibrinogen content in LIC patients. However, clues to solve this proposition may be the half-life of each coagulation factor and the dynamics of FXIII in the circulation. Fibrinogen and FXIII have long half-lives (3~4 days, 9~14 days, respectively) while Factors VIII, IX, and VII have very short half-lives, ranging from six hours to half a day [23]. The half-life of each coagulation factor invertedly parallels with its turnover rate. Additionally, all FXIII A2-B2 bind to fibrinogen in the circulation [24], suggesting that FXIII could be decreased when fibrinogen is consumed. Production and consumption of coagulation factors are occurring simultaneously in the circulation of LIC patients. Since production and consumption are similar, FXIII and fibrinogen levels could correlate well.

The trough target of FXIII for prophylaxis can be set to 10–20% in the congenital FXIII deficiency [8,9]. However, in major surgery or active bleeding, considerable amount of FXIII can be consumed locally. Randomized, placebo-controlled clinical trials in surgical patients clearly showed that FXIII supplementation reduced blood loss and increased clot firmness [25,26]. In response to these reports, the guidelines from the European Society of Anaesthesiology (ESA) suggest that FXIII concentrate (30 IU/kg) can be administered in cases of significant FXIII deficiency (i.e., < 60% activity) [27]. The daily production of factor XIII must be small as described above, suggesting that anti-coagulant therapy does not rapidly increase the intact FXIII activity in the circulation as shown in Case 1. Fibrinogen administration in case 1 or tranexamic acid alone in case 2 was insufficient, and administration of FXIII concentrates could offer favorable effects for hemostasis. Thus, FXIII supplementation can be considered for patients with LIC undergoing emergency surgery or experiencing bleeding. However, the role of FXIII supplementation in acquired FXIII deficiency remains controversial. The International Society on Thrombosis and Haemostasis specifically recommends against the use of FXIII concentrates for the management of perioperative bleeding, as the clinical benefits remain unproven [11].

In summary, our data can provide clues that testing for FXIII activity should be considered in patients with LIC, especially when such patients are scheduled to undergo any invasive procedure. However, this study has limitations due to the small sample size and the non-strict selection criteria, suggesting that the statistical analyses are exploratory and hypothesis-generating, and that our findings should be confirmed in larger, prospective studies with strict selection criteria. Evidence regarding the necessity of factor XIII administration in LIC patients also needs to be accumulated.

## Acknowledgments

We are grateful to Ms. Yukiji Ando for their secretarial support.

## Author contributions

**Conceptualization:** Takayuki Nakayama.

**Data curation:** Hiroaki Otsubo, Tsuguaki Terashima, Mai Terashima, Mika Ogawa, Hidefumi Kato, Shohei Mizuno, Satsuki Murakami, Akiyoshi Takakmi, Katsuhiko Matsuyama, Hiroshi Furukawa, Eizo Watanabe, Takayuki Nakayama.

**Formal analysis:** Siqiang Gao.

**Funding acquisition:** Takayuki Nakayama.

**Investigation:** Takayuki Nakayama.

**Methodology:** Takayuki Nakayama.

**Supervision:** Kyosuke Takeshita, Takayuki Nakayama.

**Validation:** Takayuki Nakayama.

**Writing – original draft:** Takayuki Nakayama.

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
