## [Decision Letter · Decision Letter 0]

10 Nov 2025

Dear Dr. Nakayama,

We look forward to receiving your revised manuscript.

Kind regards,

Mehmet Baysal

Academic Editor

PLOS ONE

Journal Requirements:

https://journals.plos.org/plosone/s/file?id=ba62/PLOSOne_formatting_sample_title_authors_affiliations.pdf....

“JSPS KAKENHI JB25K11657”

3. We note that your Data Availability Statement is currently as follows: [All relevant data are within the manuscript.]

Reviewers' comments:

Reviewer's Responses to Questions

**Comments to the Author**

1. Is the manuscript technically sound, and do the data support the conclusions?

Reviewer #1: Partly

Reviewer #2: Partly

Reviewer #3: Partly

Reviewer #4: Partly

2. Has the statistical analysis been performed appropriately and rigorously?

Reviewer #1: Yes

Reviewer #2: No

Reviewer #3: Yes

Reviewer #4: Yes

3. Have the authors made all data underlying the findings in their manuscript fully available?

Reviewer #1: Yes

Reviewer #2: Yes

Reviewer #3: Yes

Reviewer #4: Yes

4. Is the manuscript presented in an intelligible fashion and written in standard English?

Reviewer #1: No

Reviewer #2: Yes

Reviewer #3: Yes

Reviewer #4: Yes

Reviewer #1: Overall impression:

• The manuscript addresses an important and novel topic.

• Major revisions are required for clarity, structure, and reproducibility, particularly in Methods and Results.

• Minor edits for grammar, terminology, and flow will enhance readability and impact.

• After revisions, the manuscript could provide a valuable contribution regarding FXIII deficiency in LIC and its clinical implications.

Suggestions:

Major

• Introduction lines 69-74: “These facts led us to survey the activity of FXIII … but the decrease in FXIII was limited”. Information provided is inappropriate for an Introduction and should rather feature under Results. The final paragraph should explicitly state the study aim/s and highlight how this study addresses the identified gap.

• Methods section: The paragraph is dense and difficult to follow. Suggest dividing it into paragraphs or subsections to improve clarity.

• Methods lines 76-79: This initial sentence is difficult to read. Suggest “Between July 2017 and March 2023, clinicians retrospectively identified patients with vascular anomalies or traumatic injuries who presented with bleeding symptoms and/or abnormal coagulation markers (e.g., elevated FDP)”

• Methods line 80: Explicitly state what the second set of dates were used for. Suggest: “Clinical records were accessed for research purposes between October 3, 2024, and December 11, 2024.”

• Methods section: This section is incomplete with the following details missing:

o Clear inclusion criteria (define abnormal coagulation markers; It is not clear whether both patients with vascular anomalies and those with traumatic injuries were included in the same cohort, or whether trauma patients served as a comparison group)

o Clearly indicate if whether exclusion criteria were applied

o Define “above a certain size” - line 86

o Clarify the timing of DIC scoring and FXIII levels

o Define how the following will be identified:

Coagulopathy and chronic coagulopathy

bleeding tendency

fibrinolytic phenotype

• Statistical analysis section: This section is incomplete and does not cover all statistical analyses performed. Additional, statistical significance is not defined.

• Results section: Ensure statistical reporting is consistent (R values and P values for all correlations, where possible).

• Discussion section: Link the cases under Results to specific points in the Discussion.

• TABLE 1: Review abbreviations list in legend - AAA is incorrectly written in full as “thoracic aortic aneurysm”.

Minor

• Abstract line 30: Review the use of “;”. Suggest removing “;” to read “We collected plasma from 17 patients with LIC (9 with aneurysms and 8 with vessel malformations)…”

• Abstract line 32: suggest replacing “..measured FXIII contents” with “..measured FXIII levels”.

• Abstract line 33: suggest removing “actually” from “..13 of 17 patients actually showed bleeding tendency …”.

• Abstract line 41: Grammar – add “s” to “level”. Should be “… considered in LIC patients even with normal fibrinogen levels, who are…”

• Introduction line 59-60: regarding “However, LIC patients with normal levels of platelets and fibrinogen can be hemorrhagic for unknown reasons”, add a reference or clearly state that this is a personal clinical observation.

• Results lines 105 and 108: “..decreased (Table 1; 7 to 59%, mean 33.1%)”. Suggest revision of how results are written. Suggest “…decreased (mean 33.1%, range 7 to 59%) as shown in Table 1”. Review similar for line 108.

• Results line 114: “fibrinogen content” should rather be “fibrinogen level”

• Discussion section: Some sentences are overtly long and for readability, suggest long sentences are broken down to 2-3 sentences

Reviewer #2: The manuscript from Otsubo et al examines FXIII levels in patients with localized intravascular coagulopathy (LIC) vs those with trauma induced coagulopathy (TIC). The authors suggest that LIC is marked by profound FXIII reduction whereas TIC is not and that the level of FXIII in LIC patients positively correlated with circulating fibrinogen levels, but not other markers of coagulation factor levels and function. Two brief case reports are also presented. The strength of the manuscript is that this is one of the first to examine FXIII levels as a possibly being linked to bleeding in LIC. The weaknesses of the manuscript is the limited N number of patients in the cohorts, overstating of some conclusions, and that the case reports really provide nothing substantive to the report. I have several specific comments for authors to consider.

- The authors claim that “FXIII levels in plasma obtained from all 17 patients were significantly decreased.” However, 4 of the 17 patients have a reduction of FXIII less than 50%. Should that be considered a significant reduction? This reviewer is skeptical.

- The authors claim that TIC patients “FXIII levels were only slightly decreased”. However, but there is significant overlap in the data between LIC and TIC, thus the conclusions drawn between the two groups appears to be overstated.

- No statistical analyses are performed to attempt to document bona fide differences in FXIII between the 2 groups. This is likely due to limited N numbers of patients in each group. Additional N should be evaluated to draw definitive conclusions.

- For Figure 1, the different graphs should be labeled A, B, C, etc. In addition, P values and R values for each analysis should be displayed on the graphs.

- The strongest correlation is between FXIII levels and fibrinogen levels. However, this is not surprising as effectively all FXIII A2-B2 circulates bound to fibrinogen. If fibrinogen is lost/consumed FXIII will be as well. This should be noted.

- The case presentations add nothing to this report as they are anecdotal and fail to individually support the conclusions of the authors. In case 1 both fibrinogen and FXIII is administered, thus it is difficult to know whether it was FXIII specifically that restored hemostatic potential. For case 2, FXIII was clearly administered but were other therapies also given that could improve in neurological symptoms documented. More information is needed to determine whether this case is an example of FXIII conferring a benefit or not.

Reviewer #3: The authors describe a case series of 17 patients with localized intravascular coagulopathy and investigate the role of Factor XIII in these disorders. The authors find decreased FXIII levels in these patients. They also find that FXIII and fibrinogen levels correlate in LIC patients.

While the manuscript is overall interesting, it is very descriptive and needs further work:

Line 62: Please explain the role of FXIII in the coagulation cascade in more detail, especially its interaction with fibrinogen

Line 92: Did you assess other coagulation factors, Willebrand factor and platelet function?

Table 1: thoracic instead of thorasic, also please specify all abbreviations you have used in this table. It might be helpful to also show normal ranges for each assessed lab value.

Line 132: please list the amount per kg of FXIII.

Line 150: you are referring to several case reports, but you are only citing one reference here.

Line 165: please go into more detail why there might be a correlation between FXIII and fibrinogen

Reviewer #4: In general, this manuscript describes FXIII activity levels in patients with localized intravascular coagulopathy (LIC, n=17) and those with trauma-induced coagulopathy (TIC, n=7), along with a description of their bleeding tendencies and other coagulation parameters. The results show that FXIII activity is critically decreased in patients with LIC, but not in those with TIC. The authors also report a new correlation between FXIII activity and fibrinogen levels in LIC patients.

These findings represent the main contribution of the manuscript and constitute a novel and valuable observation that merits publication.

In the second part of the manuscript, the authors provide a more detailed description of two LIC clinical cases. FXIII concentrate was administered, in addition to other hemostatic products, due to intraoperative excessive bleeding (case 1) and chronic subdural hematoma (case 2), with good bleeding control in both cases. Based on these observations, the author’s state that “FXIII supplementation must also be required for patients with LIC undergoing emergency surgery or experiencing bleeding” (lines 196–198 of the Discussion).

However, these conclusions cannot be supported based on the observation of only two clinical cases. In particular, in case 1, the patient presented several coexisting factors that could have contributed to bleeding (low fibrinogen levels and thrombocytopenia). In case 2, the etiology of the chronic subdural hematoma is not clearly explained—was it spontaneous or traumatic? Furthermore, in case 2, the use of FXIII supplementation is highly debatable, since the FXIII level (41%) is above the minimal hemostatic threshold.

The description of these two clinical cases represents the weakest section of the manuscript and requires revision.

More detailed revision is below, and is divided into two sections: Major Revisions and Minor Revisions.

MAJOR REVISIONS

1. FXIII supplementation as a hemostatic agent

The main and most controversial issue concerns the recommendation of FXIII supplementation. The manuscript does not provide sufficient evidence to justify recommending FXIII supplementation for bleeding management in LIC patients. The two clinical cases described involve multiple bleeding risk factors, and while bleeding improvement following FXIII administration may suggest a possible effect, this interpretation remains highly questionable. In particular, in case 2, an FXIII level of 41% cannot be considered indicative of a deficiency requiring replacement therapy.

Therefore, the conclusion stated in lines 196–198 should be revised.

It is important to note that the role of FXIII supplementation in acquired FXIII deficiency remains controversial. The International Society on Thrombosis and Haemostasis (ISTH) specifically recommends against the use of FXIII concentrates for the management of perioperative bleeding, as the benefits are unproven in acquired FXIII deficiency contexts [Godier A, Greinacher A, Faraoni D, Levy JH, Samama CM. Use of factor concentrates for the management of perioperative bleeding: guidance from the SSC of the ISTH. J Thromb Haemost. 2018;16(1):170–174].

The inclusion of this reference in the bibliography is recommended.

2. Title

The reference to “FXIII supplementation” in the title should be removed, as the manuscript does not present sufficient data to support its use. The title should instead emphasize the main findings—the differences in FXIII levels between LIC and TIC patients.

3. Introduction

The section addressing acquired FXIII deficiency (lines 65–68) should be expanded. A more comprehensive discussion of the literature is needed, including information on conditions in which FXIII supplementation has been proposed, the FXIII activity thresholds that justify supplementation, and the minimal hemostatic levels reported in the literature.

4. Patients and Methods

The method used to assess FXIII activity should be clearly stated.

MINOR REVISIONS

1. Patients and Methods: When referring to DIC scores, it is recommended to include more recent references (see below).

Suggested references:

• Wada H, Thachil J, Di Nisio M, et al. Guidance for diagnosis and treatment of DIC from harmonization of the recommendations from three guidelines. J Thromb Haemost 2013; 11: 761-767.

• Levi M, Toh CH, Thachil J, Watson HG. Guidelines for the diagnosis and management of disseminated intravascular coagulation. Br J Haematol 2009; 145(01): 24-33.

2. Results: The correlation between FXIII activity and coagulation parameters in patients with TIC should be addressed, since this group serves as the control population being compared with LIC patients.

3. It is suggested that the table titles be improved for greater clarity.

Suggestion: “Clinical and laboratory profiles of patients with localized intravascular coagulopathy/trauma-induced coagulopathy”

.

Reviewer #1: No

Reviewer #2: No

Reviewer #3: No

Reviewer #4: No

---

## [Author Response · Author response to Decision Letter 1]

5 Jan 2026

December 25, 2025

Mehmet Baysal

RE: Manuscript Number: PONE-D-25-44108

Dear Dr. Baysal:

We wish to thank you and the expert reviewers for evaluating our paper entitled ““Low FXIII activity levels in patients with vascular anomalies cause bleeding tendency: FXIII supplementation as a hemostatic procedure” by Hiroaki Otsubo and Takayuki Nakayama et al. We also thank you for the invitation to submit a revised manuscript. We have addressed the comments of the reviewer properly in the revised manuscript. Thus, we now believe that the manuscript is ready for publication. If published, the manuscript will be of interest to a wide spectrum of readers of the Journal, including surgeons, physicians and researchers with interests in disseminated intravascular coagulation (DIC), perioperative managements and coagulation factors etc.

Our response in red to each of the comments of the reviewer is attached. The modifications to the manuscript are tracked in yellow and referenced in our response to the reviewers.

The corresponding author for this manuscript is Dr. Takayuki Nakayama, Department of clinical laboratory, Aichi Medical University, 1-1 Karimata, Yazako, Nagakute, Aichi 480-1195, Japan. Phone: +81-561-63-1396; FAX: +81-561-63-1396; E-mail: tnaka@aichi-med-u.ac.jp.

Thank you for considering our revised manuscript.

Sincerely yours,

Hiroaki Otsubo and Takayuki Nakayama

Reviewer #1:

We thank the referee for thoughtfully reviewing our manuscript and for suggesting ways to improve it. We are pleased to learn that the referee found that our study addressed an important and novel topic.

• The manuscript addresses an important and novel topic.

• Major revisions are required for clarity, structure, and reproducibility, particularly in Methods and Results.

• Minor edits for grammar, terminology, and flow will enhance readability and impact.

• After revisions, the manuscript could provide a valuable contribution regarding FXIII deficiency in LIC and its clinical implications.

Suggestions:

Major

• Introduction lines 69-74: “These facts led us to survey the activity of FXIII … but the decrease in FXIII was limited”. Information provided is inappropriate for an Introduction and should rather feature under Results. The final paragraph should explicitly state the study aim/s and highlight how this study addresses the identified gap.

We replaced the final paragraph with “We found that the specifically decreased Factor XIII could be involved in the bleeding tendency of LIC patients. Moreover, we describe two cases where FXIII replacement may have contributed to hemostasis in LIC patients”(line 83-84).

• Methods section: The paragraph is dense and difficult to follow. Suggest dividing it into paragraphs or subsections to improve clarity.

We divided the methods section into several subsections, as the reviewer suggested (lines 86-119).

• Methods lines 76-79: This initial sentence is difficult to read. Suggest “Between July 2017 and March 2023, clinicians retrospectively identified patients with vascular anomalies or traumatic injuries who presented with bleeding symptoms and/or abnormal coagulation markers (e.g., elevated FDP)”

As the reviewer suggested, we corrected the sentence (lines 86-90).

• Methods line 80: Explicitly state what the second set of dates were used for. Suggest: “Clinical records were accessed for research purposes between October 3, 2024, and December 11, 2024.”

As the reviewer suggested, we corrected the sentence (lines 90-91).

• Methods section: This section is incomplete with the following details missing:

o Clear inclusion criteria (define abnormal coagulation markers; It is not clear whether both patients with vascular anomalies and those with traumatic injuries were included in the same cohort, or whether trauma patients served as a comparison group)

As the reviewer requested, we clarified the inclusion criteria (line 86-88, 89-90).

o Clearly indicate if whether exclusion criteria were applied

As the reviewer requested, we clarified the exclusion criteria (line 88-89).

o Define “above a certain size” - line 86

We inserted the explanation “ which required hospitalization for intensive care” (line 90).

o Clarify the timing of DIC scoring and FXIII levels

DIC scores and FXIII levels at the initial visit were adopted (line 125).

o Define how the following will be identified:

Coagulopathy and chronic coagulopathy

We defined coagulopathy and chronic coagulopathy as FDP >20 μg/ml , FDP >20 μg/mL for more than 3 months, respectively (line 126-127).

bleeding tendency

We defined bleeding tendency as spontaneous bleeding (e.g. epistaxis) and/or continuous bleeding from wounds (line 127-129).

fibrinolytic phenotype

We defined fibrinolytic phenotype as PIC≧10 μg/ml and high FDP/D-dimer ratio ((line 135).

• Statistical analysis section: This section is incomplete and does not cover all statistical analyses performed. Additional, statistical significance is not defined.

We inserted the sentences “Statistical significance of group differences was evaluated by the Student t test using Excel software (Microsoft, Redmond, WA)” in Statistical analysis section (line 117-119), and “Statistical analysis showed that FXIII levels in LIC patients were significantly lower than those in TIC patients (P<0.05)” in the result section (line 139-140).

• Results section: Ensure statistical reporting is consistent (R values and P values for all correlations, where possible).

We added p values in APTT, PT, FDP and D-dimer (line 147-148).

• Discussion section: Link the cases under Results to specific points in the Discussion.

We inserted the sentences “tranexamic acid alone in case 2 was insufficient, and administration of FXIII concentrates could offer favorable effects for hemostasis.” in the Discussion section (line 235-236).

• TABLE 1: Review abbreviations list in legend - AAA is incorrectly written in full as “thoracic aortic aneurysm”.

We corrected the error.

Minor

• Abstract line 30: Review the use of “;”. Suggest removing “;” to read “We collected plasma from 17 patients with LIC (9 with aneurysms and 8 with vessel malformations)…”

We amended as the reviewer suggested (line 30).

• Abstract line 32: suggest replacing “..measured FXIII contents” with “..measured FXIII levels”.

We changed the wording as the reviewer suggested (line 32).

• Abstract line 33: suggest removing “actually” from “..13 of 17 patients actually showed bleeding tendency …”.

We deleted “ actually” as the reviewer suggested (line 34).

• Abstract line 41: Grammar – add “s” to “level”. Should be “… considered in LIC patients even with normal fibrinogen levels, who are…”

We added “s” as the reviewer suggested (line 41).

• Introduction line 59-60: regarding “However, LIC patients with normal levels of platelets and fibrinogen can be hemorrhagic for unknown reasons”, add a reference or clearly state that this is a personal clinical observation.

We added the sentence: according to clinical observations among physicians at our facility (line 60-61).

• Results lines 105 and 108: “..decreased (Table 1; 7 to 59%, mean 33.1%)”. Suggest revision of how results are written. Suggest “…decreased (mean 33.1%, range 7 to 59%) as shown in Table 1”. Review similar for line 108.

We have changed the order of the words as the reviewer suggested (line 131-132).

• Results line 114: “fibrinogen content” should rather be “fibrinogen level”

We have changed the wording as the reviewer suggested (line 146).

• Discussion section: Some sentences are overtly long and for readability, suggest long sentences are broken down to 2-3 sentences

We made the Discussion section simple and comprehensive as the reviewer suggested.

Reviewer #2:

We are grateful for the useful comments and suggestions that have helped us to improve our paper. We have taken all these comments and suggestions into account in the revised version of our manuscript.

The manuscript from Otsubo et al examines FXIII levels in patients with localized intravascular coagulopathy (LIC) vs those with trauma induced coagulopathy (TIC). The authors suggest that LIC is marked by profound FXIII reduction whereas TIC is not and that the level of FXIII in LIC patients positively correlated with circulating fibrinogen levels, but not other markers of coagulation factor levels and function. Two brief case reports are also presented. The strength of the manuscript is that this is one of the first to examine FXIII levels as a possibly being linked to bleeding in LIC. The weaknesses of the manuscript is the limited N number of patients in the cohorts, overstating of some conclusions, and that the case reports really provide nothing substantive to the report. I have several specific comments for authors to consider.

- The authors claim that “FXIII levels in plasma obtained from all 17 patients were significantly decreased.” However, 4 of the 17 patients have a reduction of FXIII less than 50%. Should that be considered a significant reduction? This reviewer is skeptical.

As the reviewer suggested, we replaced “significant reduction” with “mildly to severely decreased” (line 131). Not a few physicians at our facility think the decrease of FXIII in the 50% range as clinically relevant because normal range of FXIII is 70-140 %.

- The authors claim that TIC patients “FXIII levels were only slightly decreased”. However, but there is significant overlap in the data between LIC and TIC, thus the conclusions drawn between the two groups appears to be overstated.

- No statistical analyses are performed to attempt to document bona fide differences in FXIII between the 2 groups. This is likely due to limited N numbers of patients in each group. Additional N should be evaluated to draw definitive conclusions.

As the reviewer suggested, we performed statistical analysis and found that XIII levels in LIC were significantly decreased compared to those in TIC (line 139-140). We wanted to add additional data in the manuscript, but this trial has already concluded and, we are about to launch a new large-scale, prospective clinical trial to evaluate FXIII levels in LIC patients.

- For Figure 1, the different graphs should be labeled A, B, C, etc. In addition, P values and R values for each analysis should be displayed on the graphs.

As the reviewer suggested, we added labelling of graphs, P values and R values on the graphs.

- The strongest correlation is between FXIII levels and fibrinogen levels. However, this is not surprising as effectively all FXIII A2-B2 circulates bound to fibrinogen. If fibrinogen is lost/consumed FXIII will be as well. This should be noted.

We thank the reviewer for good suggestions, we added the sentence and quote a reference (line 220-224).

- The case presentations add nothing to this report as they are anecdotal and fail to individually support the conclusions of the authors. In case 1 both fibrinogen and FXIII is administered, thus it is difficult to know whether it was FXIII specifically that restored hemostatic potential. For case 2, FXIII was clearly administered but were other therapies also given that could improve in neurological symptoms documented. More information is needed to determine whether this case is an example of FXIII conferring a benefit or not.

The reviewer is correct. We had not administered fibrinogen and FXIII concentrates simultaneously in case 1. We added information that after confirming the ineffectiveness of fibrinogen administration alone, factor XIII was administered in the Case presentations section (line 168-169). We concluded that FXIII possessed favorable effects in both cases in the Discussion section (line 235-236).

Reviewer #3:

We thank the referee for thoughtfully reviewing our manuscript and for encouraging us to do a lot of works to improve our manuscript. We are pleased to learn that the referee found that our study was overall interesting.

The authors describe a case series of 17 patients with localized intravascular coagulopathy and investigate the role of Factor XIII in these disorders. The authors find decreased FXIII levels in these patients. They also find that FXIII and fibrinogen levels correlate in LIC patients.

While the manuscript is overall interesting, it is very descriptive and needs further work:

Line 62: Please explain the role of FXIII in the coagulation cascade in more detail, especially its interaction with fibrinogen

As the reviewer suggested, we add explanation regarding FXIII involvement in in fibrin clot stabilization (line 67-71).

Line 92: Did you assess other coagulation factors, Willebrand factor and platelet function?

We had evaluated activities of von Willebrand factor in most of LIC patients (11 out of 17) and found that they were within normal limits (mean 156.5%, range 63 to 300%) . We added this information in the manuscript (line 140-142). However, we had not checked the platelet functions because the systematic review already reported that platelets in patients with DIC were generally activated (Platelets. 2018 May;29(3):238-248).

Table 1: thoracic instead of thorasic, also please specify all abbreviations you have used in this table. It might be helpful to also show normal ranges for each assessed lab value.

As the reviewer suggested, we corrected Table 1 and 2.

Line 132: please list the amount per kg of FXIII.

We described the amount per kg of FXIII in patient 1 and 2 (line 170 and 182).

Line 150: you are referring to several case reports, but you are only citing one reference here.

We were not able to quote other reports because they were written in Japanese. So, we replaced “several case reports” with “a few case reports”.

Line 165: please go into more detail why there might be a correlation between FXIII and fibrinogen

As the reviewer suggested, we have added considerations regarding the dynamics of FXIII in the circulation (line 220-224).

Reviewer #4:

We thank the reviewer 4 for thoughtfully reviewing our manuscript and for suggesting ways to improve it. We are pleased to learn that the reviewer 4 found that our study contained valuable observation with merits of publication.

In general, this manuscript describes FXIII activity levels in patients with localized intravascular coagulopathy (LIC, n=17) and those with trauma-induced coagulopathy (TIC, n=7), along with a description of their bleeding tendencies and other coagulation parameters. The results show that FXIII activity is critically decreased in patients with LIC, but not in those with TIC. The authors also report a new correlation between FXIII activity and fibrinogen levels in LIC patients.

These findings represent the main contribution of the manuscript and constitute a novel and valuable observation that merits publication.

In the second part of the manuscript, the authors provide a more detailed description of two LIC clinical cases. FXIII concentrate was administered, in addition to other hemostatic products, due to intraoperative excessive bleeding (case 1) and chronic subdural hematoma (case 2), with good bleeding control in both cases. Based on these observations, the author’s state that “FXIII supplementation must also be required for patients with LIC undergoing emergency surgery or experiencing bleeding” (lines 196–198 of the Discussion).

However, these conclusions cannot be supported based on the observation of only two clinical cases. In particular, in case 1, the patient presented several coexisting factors that could have contributed to bleeding (low fibrinogen levels and thrombocytopenia). In case 2, the etiology of the chronic subdural hematoma is not clearly explained—was it spontaneous or traumatic? Furthermore, in case 2, the use of FXIII supplementation is highly debatable, since the FXIII level (41%) is above the minimal hemostatic threshold.

The description of these two clinical cases represents the weakest section of the manuscript and requires revision.

We added the traumatic etiology of the chronic subdural hematoma in the Case 2 (line 174-175).

More detailed revision is below, and is divided into two sections: Major Revisions and Minor Revisions.

MAJOR REVISIONS

1.

---

## [Decision Letter · Decision Letter 1]

28 Jan 2026

Dear Dr. Nakayama,

plosone@plos.org. . . . A letter that responds to each point raised by the academic editor and reviewer(s). You should upload this letter as a separate file labeled 'Response to Reviewers'.A marked-up copy of your manuscript that highlights changes made to the original version. You should upload this as a separate file labeled 'Revised Manuscript with Track Changes'.An unmarked version of your revised paper without tracked changes. You should upload this as a separate file labeled 'Manuscript'.

We look forward to receiving your revised manuscript.

Kind regards,

Mehmet Baysal

Academic Editor

PLOS One

Journal Requirements:

Reviewers' comments:

Reviewer's Responses to Questions

**Comments to the Author**

Reviewer #1: All comments have been addressed

Reviewer #2: (No Response)

Reviewer #3: All comments have been addressed

Reviewer #4: All comments have been addressed

2. Is the manuscript technically sound, and do the data support the conclusions?

Reviewer #1: Yes

Reviewer #2: No

Reviewer #3: Yes

Reviewer #4: Yes

3. Has the statistical analysis been performed appropriately and rigorously?

Reviewer #1: Yes

Reviewer #2: No

Reviewer #3: Yes

Reviewer #4: Yes

4. Have the authors made all data underlying the findings in their manuscript fully available?

Reviewer #1: Yes

Reviewer #2: Yes

Reviewer #3: Yes

Reviewer #4: Yes

5. Is the manuscript presented in an intelligible fashion and written in standard English?

Reviewer #1: Yes

Reviewer #2: Yes

Reviewer #3: Yes

Reviewer #4: Yes

Reviewer #1: Some minor comments:

- Suggest the study aim/s is/are clearly stated in a sentence in the final paragraph of the Introduction rather than just summarizing the findings.

-Given the small sample size, suggest adding a sentence indicating that the statistical analyses are exploratory and hypothesis-generating.

Reviewer #2: The authors were partially responsive to critiques from review of the original manuscript. Minor changes were made to the manuscript but the fatal flaws still exist. The study is significantly underpowered to draw meaningful conclusions and the case reports continue to be anecdotal.

For the statistical analysis of the FXIII levels in LIC vs TIC patients, the authors used a student T-test. This relies on a normal distribution of the data. The authors should have performed normality testing to determine if the T-test was valid as opposed to a nonparametric analysis.

Reviewer #3: Thanks for changing the manuscript, which has improved its overall quality. For the p-values, I would appreciate a tiered report with an additional description in the methods, such as statistically significant = <0.05, reporting p-values ≥0.05 to 0.1, and writing n.s. for p-values ≥0.1. Thanks.

Reviewer #4: General Comments

The authors have substantially revised the manuscript and have addressed most of the reviewer’s comments. However, some minor issues and corrections remain, mainly within the Discussion section, in order to clarify specific points related to FXIII levels and FXIII supplementation.

These issues are detailed below.

1. Introduction

1.1. In Major Comments, point 3, the reviewer requested that the minimal hemostatic levels of FXIII reported in the literature be stated. This information is still missing in the revised version of the manuscript.

It is recommended that this data be added at the end of line 80.

1.2. Lines 70–71: There is a repetition of words that should be corrected.

2. Patients and Methods

Blood Testing

It should be stated that some patients also underwent measurements of von Willebrand factor (vWF), as this parameter is mentioned in the Results section (lines 140–142). Please specify which test was used (e.g., vWF activity, vWF antigen [vWF:Ag], vWF ristocetin cofactor activity [vWF:RCo], or another assay).

In line 140, the phrase “activities of von Willebrand factor” should be corrected to either “von Willebrand factor activity” or “von Willebrand factor level”.

3. Results

The sentences from lines 152 to 155 are unclear due to issues with the English language. Please revise this paragraph to improve clarity and readability.

4. Discussion

Several deficiencies remain in the Discussion section.

4.1. The sentence in lines 190–192:

“Kleber et al. suggested that a cut-off for FXIII activity of <50–70% may be appropriate to diagnose acquired FXIII deficiency, although opinions and protocols may vary [16].”

Should be revised to:

Kleber et al. suggested that a cut-off for FXIII activity of <50–70% may be appropriate to diagnose acquired FXIII deficiency; however, this range is very broad, and opinions and diagnostic protocols may vary [16].

4.2. The sentence in lines 193–195:

“Here, we showed that FXIII was considerably decreased in all cases with vascular anomalies (Table 1; mean 33.1%, range 7–59%) while acquired FXIII deficiency was limited in TIC patients (Table 2; mean 67.4%, range 39–98%), as reported previously.”

Should be revised to:

Here, we showed that FXIII was decreased in some cases with vascular anomalies (Table 1; mean 33.1%, range 7–59%), whereas acquired FXIII deficiency was not present in TIC patients (Table 2; mean 67.4%, range 39–98%), consistent with previous reports.

4.3. Line 198: Please remove the word “substantial”.

4.4. Lines 209–213:

Instead of:

“In TIC patients, FXIII levels were most strongly correlated with APTT (R = −0.884, P < 0.05) and PT-INR (R = −0.798, P < 0.05), as described above. These results suggest that LIC and TIC show similar fibrinolytic phenotypes, but the underlying mechanism is quite different.”

Use:

In TIC patients, FXIII levels did not correlate with fibrinogen levels and were most strongly correlated with APTT (R = −0.884, P < 0.05) and PT-INR (R = −0.798, P < 0.05), as described above. These findings suggest that LIC and TIC may exhibit similar fibrinolytic phenotypes; however, the underlying pathogenic mechanisms are quite different.

4.5. Line 214: Please remove the expression “as to”.

4.6. Line 215: Please remove the word “the” in the phrase “the proposition”.

4.7. Lines 238–239:

The following sentence should be divided into two sentences:

“However, the role of FXIII supplementation in acquired FXIII deficiency remains controversial as the International Society on of Thrombosis and Haemostasis specifically recommends against the use of FXIII concentrates for the management of perioperative bleeding because the benefits are still unproven [11].”

Suggested revision:

However, the role of FXIII supplementation in acquired FXIII deficiency remains controversial. The International Society on Thrombosis and Haemostasis specifically recommends against the use of FXIII concentrates for the management of perioperative bleeding, as the clinical benefits remain unproven [11].

.

Reviewer #1: No

Reviewer #2: No

Reviewer #3: No

Reviewer #4: No

---

## [Author Response · Author response to Decision Letter 2]

3 Mar 2026

Academic Editor:

The authors must address these shortcomings to improve the study's validity. Specifically, the small sample size limits the statistical power and generalizability of the findings. Additionally, failing to test for normality (e.g., using Shapiro-Wilk or Kolmogorov-Smirnov tests) risks the use of inappropriate parametric tests on non-normal data, potentially skewing results.For the futer the authors shoulde also recommend future multi-center studies to achieve a larger sample size.

We have inserted additional descriptions about the limitations of this study due to the small sample size and the non-strict selection criteria as described below. We also have performed Shapiro-Wilk tests, showing that P values were 0.7508 in the LIC subgroup and 0.6285 in the TIC subgroup. We inserted this information in the Method and Result section. (lines 122-123, lines 148-151)

Reviewer #1:

- Suggest the study aim/s is/are clearly stated in a sentence in the final paragraph of the Introduction rather than just summarizing the findings.

We have inserted the sentence “These facts led us to survey the activity of FXIII, which can be involved in delayed wound healing and bleeding tendency, in the plasma of patients with LIC” as the reviewer suggested. (lines 82-83)

-Given the small sample size, suggest adding a sentence indicating that the statistical analyses are exploratory and hypothesis-generating.

We have inserted the sentence “the statistical analyses are exploratory and hypothesis-generating” in the discussion section as the reviewer suggested. (lines 258-259)

Reviewer #2:

The authors were partially responsive to critiques from review of the original manuscript. Minor changes were made to the manuscript but the fatal flaws still exist. The study is significantly underpowered to draw meaningful conclusions and the case reports continue to be anecdotal.

We deeply appreciate the reviewer for meaningful suggestions. We are trying hard to perform a larger prospective trials as a next step, which will reveal whether our findings are true or not.

For the statistical analysis of the FXIII levels in LIC vs TIC patients, the authors used a student T-test. This relies on a normal distribution of the data. The authors should have performed normality testing to determine if the T-test was valid as opposed to a nonparametric analysis.

We have performed Shapiro-Wilk tests, showing that P values were 0.7508 in the LIC subgroup and 0.6285 in the TIC subgroup. We inserted this information in the Method and Result section. (lines 122-123, lines 148-151)

Reviewer #3:

Thanks for changing the manuscript, which has improved its overall quality. For the p-values, I would appreciate a tiered report with an additional description in the methods, such as statistically significant = <0.05, reporting p-values ≥0.05 to 0.1, and writing n.s. for p-values ≥0.1. Thanks.

We inserted the sentence “P values < 0.05 and ≥0.1 were considered statistically significant and not statistically significant, respectively. When P values were marginally significant (≥0.05 to <0.1), we described P values” in the Methods and Result section, and Fig1, as the reviewer suggested.(lines 125-128, lines 163-166)

Reviewer #4:

General Comments

The authors have substantially revised the manuscript and have addressed most of the reviewer’s comments. However, some minor issues and corrections remain, mainly within the Discussion section, in order to clarify specific points related to FXIII levels and FXIII supplementation.

These issues are detailed below.

We thank the reviewer for carefully reviewing our manuscript again and for suggesting ways to improve it. We are pleased to learn that we had addressed most of the reviewer’s comments. Now, we have revised the rest.

1. Introduction

1.1. In Major Comments, point 3, the reviewer requested that the minimal hemostatic levels of FXIII reported in the literature be stated. This information is still missing in the revised version of the manuscript.

It is recommended that this data be added at the end of line 80.

We have inserted the sentence “Whereas the risk of spontaneous bleeding in congenital FXIII deficiency increases below ~10-15% activity of FXIII [12]” at the end of line 80.

1.2. Lines 70–71: There is a repetition of words that should be corrected.

We have corrected the error, as the reviewer suggested.

2. Patients and Methods

Blood Testing

It should be stated that some patients also underwent measurements of von Willebrand factor (vWF), as this parameter is mentioned in the Results section (lines 140–142). Please specify which test was used (e.g., vWF activity, vWF antigen [vWF:Ag], vWF ristocetin cofactor activity [vWF:RCo], or another assay).

We have inserted the sentence “Some patients also underwent measurements of von Willebrand factor (vWF) activity. The plasma samples were sent on the day of blood draw to a commercial laboratory (BML Inc., Tokyo, Japan) for assessment of vWF ristocetin cofactor activity” (lines 115-118, in the Method section).

In line 140, the phrase “activities of von Willebrand factor” should be corrected to either “von Willebrand factor activity” or “von Willebrand factor level”.

We have replaced “activities of von Willebrand factor” with “von Willebrand factor activity” (line 152) as the reviewer suggested.

3. Results

The sentences from lines 152 to 155 are unclear due to issues with the English language. Please revise this paragraph to improve clarity and readability.

We have revised the paragraph as “In TIC patients, FXIII activity showed a significant negative correlation with APTT (R = -0.884, P < 0.05) and PT-INR (R = -0.798, P < 0.05). No significant correlations were found with fibrinogen (R=0.297, P>0.1), platelet counts (R=0.124, P>0.1), FDP (R=-0.568, P>0.1), or D-dimer (R=-0.555, P>0.1)”. (lines 163-166)

4. Discussion

Several deficiencies remain in the Discussion section.

4.1. The sentence in lines 190–192:

“Kleber et al. suggested that a cut-off for FXIII activity of <50–70% may be appropriate to diagnose acquired FXIII deficiency, although opinions and protocols may vary [16].”

Should be revised to:

Kleber et al. suggested that a cut-off for FXIII activity of <50–70% may be appropriate to diagnose acquired FXIII deficiency; however, this range is very broad, and opinions and diagnostic protocols may vary [16].

We have inserted the sentence “however, this range is very broad, and opinions and diagnostic protocols may vary [17].” (lines 203-204)

4.2. The sentence in lines 193–195:

“Here, we showed that FXIII was considerably decreased in all cases with vascular anomalies (Table 1; mean 33.1%, range 7–59%) while acquired FXIII deficiency was limited in TIC patients (Table 2; mean 67.4%, range 39–98%), as reported previously.”

Should be revised to:

Here, we showed that FXIII was decreased in some cases with vascular anomalies (Table 1; mean 33.1%, range 7–59%), whereas acquired FXIII deficiency was not present in TIC patients (Table 2; mean 67.4%, range 39–98%), consistent with previous reports.

We have revised the sentence, as the reviewer suggested (lines 204-207).

4.3. Line 198: Please remove the word “substantial”.

We have deleted the word, as the reviewer suggested.

4.4. Lines 209–213:

Instead of:

“In TIC patients, FXIII levels were most strongly correlated with APTT (R = −0.884, P < 0.05) and PT-INR (R = −0.798, P < 0.05), as described above. These results suggest that LIC and TIC show similar fibrinolytic phenotypes, but the underlying mechanism is quite different.”

Use:

In TIC patients, FXIII levels did not correlate with fibrinogen levels and were most strongly correlated with APTT (R = −0.884, P < 0.05) and PT-INR (R = −0.798, P < 0.05), as described above. These findings suggest that LIC and TIC may exhibit similar fibrinolytic phenotypes; however, the underlying pathogenic mechanisms are quite different.

We have revised the sentence, as the reviewer suggested (lines 221-225).

4.5. Line 214: Please remove the expression “as to”.

We have deleted “as to”, as the reviewer suggested.

4.6. Line 215: Please remove the word “the” in the phrase “the proposition”.

We have deleted “the”, as the reviewer suggested.

4.7. Lines 238–239:

The following sentence should be divided into two sentences:

“However, the role of FXIII supplementation in acquired FXIII deficiency remains controversial as the International Society on of Thrombosis and Haemostasis specifically recommends against the use of FXIII concentrates for the management of perioperative bleeding because the benefits are still unproven [11].”

Suggested revision:

However, the role of FXIII supplementation in acquired FXIII deficiency remains controversial. The International Society on Thrombosis and Haemostasis specifically recommends against the use of FXIII concentrates for the management of perioperative bleeding, as the clinical benefits remain unproven [11].

We have revised the sentence, as the reviewer suggested (lines 250-254).

---

## [Decision Letter · Decision Letter 2]

24 Mar 2026

Low FXIII activity levels in patients with vascular anomalies, not in patients with trauma-induced coagulopathy

PONE-D-25-44108R2

Dear Dr. Nakayama,

We’re pleased to inform you that your manuscript has been judged scientifically suitable for publication and will be formally accepted for publication once it meets all outstanding technical requirements.

Kind regards,

Mehmet Baysal

Academic Editor

PLOS One

Additional Editor Comments (optional):

Reviewers' comments:

Reviewer's Responses to Questions

**Comments to the Author**

Reviewer #1: All comments have been addressed

Reviewer #3: All comments have been addressed

Reviewer #4: All comments have been addressed

2. Is the manuscript technically sound, and do the data support the conclusions?

Reviewer #1: Yes

Reviewer #3: Yes

Reviewer #4: Yes

3. Has the statistical analysis been performed appropriately and rigorously?

Reviewer #1: Yes

Reviewer #3: Yes

Reviewer #4: Yes

4. Have the authors made all data underlying the findings in their manuscript fully available?

Reviewer #1: Yes

Reviewer #3: Yes

Reviewer #4: Yes

5. Is the manuscript presented in an intelligible fashion and written in standard English?

Reviewer #1: Yes

Reviewer #3: Yes

Reviewer #4: No

Reviewer #1: This is a clinically relevant study, but conclusions are too strong - the data show an association between low FXIII and LIC, not that it causes bleeding, especially without comparing bleeding vs non-bleeding patients. Also the case reports cannot prove that FXIII treatment was effective. The results are hypothesis-generating and should be interpreted more cautiously. Otherwise, no further edits needed in my opinion.

Reviewer #3: The authors have addressed all issues raised by the four reviewers in the last revision. Therefore, the I accept this revision for publication.

Reviewer #4: The revised manuscript is significantly clearer and provides more detailed information, which greatly enhances the understanding of the work.

.

Reviewer #1: No

Reviewer #3: No

Reviewer #4: No

---

## [Editor Report · Acceptance letter]

PONE-D-25-44108R2

PLOS One

Dear Dr. Nakayama,

I'm pleased to inform you that your manuscript has been deemed suitable for publication in PLOS One. Congratulations! Your manuscript is now being handed over to our production team.

Kind regards,

on behalf of

Dr. Mehmet Baysal

Academic Editor

PLOS One